# The value of satellite observations in the analysis and short-range prediction of Asian dust

Angela Benedetti[1], Francesca Di Giuseppe[1], Luke Jones[1], Vincent-Henri Peuch[1], Samuel Rémy[2], and Xiaoye Zhang[3,4]

[1]European Centre for Medium Range Weather Forecast (ECMWF), Reading, UK
[2]Institut Pierre-Simon Laplace, CNRS / Sorbonne Université, Paris, France
[3]State Key Laboratory of Severe Weather & Key Laboratory of Atmospheric Chemistry of CMA, Chinese Academy of Meteorological Sciences, Beijing, China
[4]Center for Excellence in Regional Atmospheric Environment, INE, CAS

*Correspondence to:* Angela Benedetti (A.Benedetti@ecmwf.int)

**Abstract.** Asian dust is a seasonal meteorological phenomenon which affects East Asia, and has severe consequences on the air quality of China, North and South Korea and Japan. Despite the continental extent, the prediction of severe episodes and the anticipation of their consequences is challenging. Three one-year experiments were run to assess the skill of the model of the European Centre for Medium-Range Weather Forecasts (ECMWF) in monitoring Asian dust and understand its relative

contribution to the aerosol load over China. Data used were the MODIS Dark Target and the Deep Blue Aerosol Optical Depth. In particular the experiments aimed at understanding the added value of data assimilation runs over a model run without any aerosol data. The year 2013 was chosen as representative for the availability of independent Aerosol Optical Depth (AOD) data from two established ground-based networks (AERONET and CARSNET), which could be used to evaluate experiments. Particulate Matter (PM) data from the China Environmental Protection Agency were also used in the evaluation. Results show

that the assimilation of satellite AOD data is beneficial to predict the extent and magnitude of desert-dust events and to improve the short-range forecast of such events. The availability of observations from the MODIS Deep Blue algorithm over bright surfaces is an asset, allowing for a better localization of the sources and definition of the dust events. In general both experiments constrained by data assimilation perform better that the unconstrained experiment, generally showing smaller mean normalized bias and fractional gross error with respect to the independent verification datasets. The impact of the assimilated satellite

observations is larger at analysis time, but lasts into the forecast up to 48 hours. The performance of the global model in terms of particulate matter does not show the same degree of skill as in term of optical depth. Despite this, the global model is able to capture some regional pollution patterns. This indicates that the global model analyses may be used as boundary conditions for regional air quality models at higher resolution, enhancing their performance in situations when part of the pollution may have originated via large-scale mechanisms. While assimilation is not a substitute for model development and characterization

of the emission sources, results indicate that it can play a role in delivering improved monitoring of Asian dust optical depth.

# 1 Introduction

Asian dust is a seasonal meteorological phenomenon which affects East Asia, particularly during spring. Dust is transported from the deserts of Taklimakan, Mongolia, northern China and Kazakhstan and can travel over long distances affecting air quality of China, North and South Korea and Japan. Occasionally, the dust can even be transported over the ocean over thousands of kilometers and reach the coasts of North America. Measurements at Manua Loa, Hawaii show presence of dust of Asian origin (Shaw, 1980; Darzi and Winchester, 1982; Holmes and Zoller, 1996).

Dust is becoming a more serious concern due to the increase of industrial pollutants contained in it, also in connection to intensified desertification in China causing longer and more frequent occurrences, both linked to climate factors and to land use change in West Asia (Zhang et al., 2003; Chuluun and Ojima, 2002). Areas affected by dust experience poor air quality and, as a consequence, dust is known to cause a variety of health problems, including sore throat and asthma in otherwise healthy people. Often, people are advised to avoid or minimize outdoor activities, depending on the severity of the storms. For those already with asthma or respiratory infections, dust-related pollution can be fatal (Lee et al., 2013; Goudie, 2014; Chen et al., 2011). The economic impact of dust storms connected to reduced air visibility, canceled flights, and disrupted ground travel can also be high (Jeong, 2008).

Because of its health, social and economical impacts, it is critical to understand the source strength, transport and deposition of dust and to establish Sand and Dust Storms (SDS) forecasting and early warning capabilities. Under the auspices of the World Meteorological Organization (WMO), the need for a global SDS forecasting and early warning system was highlighted since 2005. More than forty member countries expressed interest in participating in activities to improve capacities for more reliable SDS monitoring, forecasting and assessment. The SDS-WAS (Warning and Assessment System) was launched in 2007 with the mission to enhance the ability of countries to deliver timely and quality sand and dust storm forecasts, observations and information to users through an international partnership of research and operational communities. Towards this goal, three regional centers have been created: the North Africa-Middle East-European (NAMEE) node, the Pan-American node and the Asian node. The latter is hosted by the Chinese Meteorological Administration (CMA) and its focus is Asian dust prediction (Zhang and coauthors, 2015). Member countries of the Asian node include Japan, Korea, Mongolia and Kazakhstan. All centres run their forecast models for the specific region (Gong and Zhang, 2008; Park et al., 2010). Some models such as that of the Japan Meteorological Agency run at the global scale (Tanaka and Chiba, 2005). The European Centre for Medium-range Weather Forecast (ECMWF) also contributes its dust forecasts to the Asian node through a partnership between the Copernicus Atmosphere Monitoring Service (CAMS) and CMA[1].

Despite the recognized relevance, Asian dust monitoring and forecasting are still a challenge due to the poor characterization of the sources, which have strong spatial and temporal variability and the paucity of observational data available to characterize both emissions and transport especially in Near-Real-Time (NRT). A study by Uno et al. (2006) compared eight state-of-the-art dust models and showed that emission fluxes from the Taklimakan Desert and Mongolia strongly differ among the models. They suggested that measurements of dust flux and accurate updated land use information are important to improve the models over

---

[1]See http://eng.nmc.cn/sds_was.asian_rc/ for further reference

these regions. They also found that modeling of dust transport and removal processes over key regions needs to be improved in order to obtain more skillful dust prediction that could underpin warning systems, improve preparedness and possibly lessen the impacts.

Although modeling problems need to be tackled at the level of the parameterization of the physical, dynamical and radiative processes, data assimilation of dust-related observations such as optical depth and lidar backscatter from satellite sensors can help alleviate problems and improve the forecast of dust events. This has been shown for example in a pioneering work by Sekiyama et al. (2010) where the authors show that assimilation of data from the Cloud-Aerosol Lidar and Infrared Pathfinder Satellite Observations (CALIPSO) lidar in an Ensemble Kalman filter framework improves the model simulation of Aeolian dust events. Assimilation efforts to specifically improve dust prediction have been undertaken at the global and regional scale by several institutes both with ensemble methods where also emissions are perturbed (Lin et al., 2008; Rubin et al., 2016; Di Tomaso et al., 2017) and variational methods (Niu et al., 2008; Liu et al., 2011). All studies show the positive impact of satellite and in-situ data assimilation in desert dust-affected regions for high load episodes, with some impact lasting into the forecast range.

Progress has been made in recent years thanks to the deployment of accurate, multi-channel aerosol sensors on satellite platforms and the improvement of ground-based networks which are now offering stable observations of good quality over long period and increasingly in NRT. Given these opportunities in this paper, we will investigate the added benefit of data assimilation for the desert dust analysis and short-range prediction using ECMWF's Integrated Forecasting System (IFS) and AOD observations from Moderate Resolution Imaging Spectro-radiometer (MODIS) which are now also available on bright surfaces (Deep blue collection) in the framework of the 4D-Var assimilation system extended to aerosol species (Courtier et al., 1994; Benedetti et al., 2009). The novelty with respect to previous studies that also employed MODIS observations (i.e. Liu et al. (2011)) rests in the use of a fully *operational* 4D-Var system in which the model dynamics serve as a constraint during the assimilation. For dust storms of synoptic extent, this constraint might imply a longer-lasting impact of the assimilated data into the forecast range. The verification of the analysis and forecast benefits from the availability of data from the Aerosol Robotic Network (AERONET, Holben et al. (1998, 2001)), as well from the China Aerosol Remote Sensing Network (CARSNET, Che et al. (2015)). Additionally, PM data from the China Environmental Protection Agency (CEPA) are also used to assess the value of AOD assimilation in surface PM forecasts.

## 2 Description of the ECMWF/CAMS system

Desert dust forecasts are provided by the ECMWF/CAMS system which is a comprehensive modeling and assimilation system based on the IFS developed at ECMWF with the extended capabilities of including atmospheric composition tracers in the transport model and in the analysis (Flemming et al., 2015). The system has been developed in the CAMS precursor projects, Global and Regional Earth System Monitoring Using Satellite and In Situ Data (GEMS, Hollingsworth et al. (2008)) and Monitoring Atmospheric Composition and Climate (MACC, Peuch and Engelen (2012)). Aerosols are forecast within the ECMWF/CAMS global system by an aerosol model (Morcrette et al., 2009), based on earlier work by Reddy et al. (2005) and

Boucher et al. (2002) that uses five species: dust, sea salt, black carbon, organic carbon and sulfates. In the version of the model used in this study, nitrates were not included. More recent versions of the model, do include this important aerosol component.

Dust aerosols specifically are represented by three prognostic variables that correspond to three size bins, with bin limits of 0.03, 0.55, 0.9 and 20 $\mu m$ radius. Specific dust processes that are parameterized are production of dust through saltation and removal by wet and dry deposition and sedimentation. Various areas are marked as having potential to emit dust based on surface albedo, moisture of the top soil level and bare soil fraction. Dust emissions are parameterized following Ginoux et al. (2001) as a function of the cubic power of 10m wind speed. A correction is also applied to account for gustiness as described in Morcrette et al. (2008). Dry deposition depends on a prescribed deposition velocity and on the aerosol concentration in the model level closer to the surface.

For sea-salt, a bin representation is used and emissions are also parameterized as a function of surface winds. For all other tropospheric aerosols (carbonaceous aerosols and sulfates), a bulk parameterization is used and emission sources are defined according to established inventories (Lamarque et al., 2010). Biomass burning emissions contributing to black carbon and organic matter loads are prescribed from the Global Fire Assimilation System (GFAS,Kaiser et al. (2012)). Removal processes include sedimentation for dust and sea salt only, and wet and dry deposition and in-cloud and below cloud scavenging for all particles. A simplified representation of the sulfur cycle is also included with only two variables, sulfur dioxide ($SO_2$) acting as a precursor for sulphate ($SO_4$) in particulate form. Overall, a total of 12 additional prognostic variables for the mass mixing ratio of the different components (bins or types) of the various aerosols are used in this configuration. The ECMWF/CAMS system also includes chemically reactive tracers such as ozone, carbon monoxide, NOx, formaldehyde as well as greenhouse gases.

The ECMWF/CAMS system is used operationally to provide global forecasts of aerosols, reactive gases and greenhouse gases as well as boundary conditions for higher resolution regional models proving air quality forecasts over Europe. Data are freely available from the CAMS website (https://atmosphere.copernicus.eu/).

## 3 Methodology and experimental set-up

The ECMWF/CAMS system also runs a 4D-Var assimilation of satellite and ground-based data for the meteorological variables as well as the atmospheric composition variables. The aerosol assimilation uses total aerosol mixing ratio as a control variable. Each aerosol species is assumed to contribute to the total aerosol mixing ratio according to a fractional contribution which is assumed constant over the 12-hour assimilation window. At the end of the minimization, the increments in total aerosol mixing ratio are distributed to the individual species according to their fractional contribution to the total mass. A forward model is used to calculate the Aerosol Optical Depth (AOD) from the individual components using a look-up table approach. The aerosol optical properties are prescribed according to Bozzo et al. (2017). During the minimization, the tabulated optical properties of the individual species are retrieved and used in the calculation of the total extinction. The vertical integral of the extinction profile provided the total AOD, which is the observed quantity, at every observation location. The vertical profile of the aerosol mixing ratio is not modified by the assimilation as only AOD is used as observation. Thus, the vertical profile is

largely dictated by the model and the distribution of the increments in the vertical is due to the vertical structure function in the background error.

AOD data at 550nm from the MODIS sensors on board of the Terra and Aqua satellites have been assimilated in the ECMWF'4D-Var system since 2008 (Benedetti et al., 2009). Initially, only the Dark Target product was used (Levy et al., 2010). Since 2015, both MODIS Dark Target and Deep Blue (Hsu et al., 2013) data are used. Deep Blue offers the advantage to provide Aerosol Optical Depth(AOD) retrievals over bright surfaces, which is of particular interest for getting insight on dust sources by means of data assimilation. It can be assumed that over deserts the biggest contribution to AOD comes from dust aerosols, even if the analysis uses total AOD and the other aerosol species are also included. Recently, MODIS Collection 6 has become available (Levy et al., 2013; Sayer et al., 2013, 2014). This is the collection that has been used in this study.

The operational ECMWF/CAMS systems also uses AOD data from the Polar Multi-Sensor Aerosol Optical Properties — Metop (PMAP) product, operationally provided by EUMETSAT after February 2014, and hence not available for the experiments outlined in this paper. Assimilation of this product for the purpose of the application outlined here would need an independent assessment.

To investigate the impact of the assimilation on the dust prediction, three experiments were set-up for the whole of 2013: one experiment with 4D-Var data assimilation of MODIS Col 6 with the exclusion of Deep Blue data (indicated hereafter as DT), one experiment with 4D-Var assimilation of MODIS Col 6 data with the inclusion of Deep Blue data (indicated as DTDB) and one control experiment with free-running aerosols (indicated as CONTROL). In the control experiment, no aerosol data are assimilated in the analysis. In this paper, we focus on results during the spring season (March-April-May 2013), when several marked events occurred.

The horizontal resolution of the operational version run in the context of CAMS is currently 40km with 60 vertical levels and model top at 0.1 hPa. The experiments shown here were run with a previous version of the system, comparable with the CAMS configuration for 2013, with horizontal resolution of 80km and the same number of vertical levels.

## 4 Outcomes of the investigation

### 4.1 Comparisons with independent AOD datasets

Results from the three experiments (DT, DTDB and CONTROL) are presented in what follows. Comparisons were made with independent datasets from AERONET and CARSNET. AERONET data are routinely used for the verification of the ECMWF/CAMS forecasts, but this is the first time that the CARSNET dataset has been used in this context. Model data was bilinearly interpolated to the AERONET/CARSNET site locations and then averaged over 24 hour periods (from T+3 to T+24). AERONET data is similarly averaged, with each data value receiving a weight proportional to the time difference between the data values before and after it, up to a maximum of three hours. The CARSNET data used was already in the form of daily averages and no further averaging was done.

While the entire year was considered, we focus in this paper on the spring season when Asian dust is at its peak due to the seasonal increased frequency of cyclonic weather systems with associated high surface wind speeds. At this time of the year

the contrast between warm air masses in the south and cold air masses from Siberia and Mongolia is maximum. The high north-south thermal gradient can give rise to situations characterized by a high baroclinic instability, with the development of fast developing and moving lows associated with high winds over the dust emitting regions of the Taklimakan and Gobi deserts and inner Mongolia.

As an example of the extent of this type of events, figure 1 shows the March 9th dust storm as simulated by the ECMWF model along with the coverage provided by the MODIS Dark Target and Deep Blue AOD retrievals. Comparing with the satellite image on the same day (figure 2), courtesy of NASA, the model shows a good degree of qualitative agreement with the observations.

    This particular dust storm was analyzed in detail by Chauhan et al. (2016). The storm originated in the Taklimakan and dust

was first transported to Northeastern China and further transported to the southeast. The air quality in Beijing was severely affected as the sandstorm spread. The dust further propagated to the Korean Peninsula and Japan.

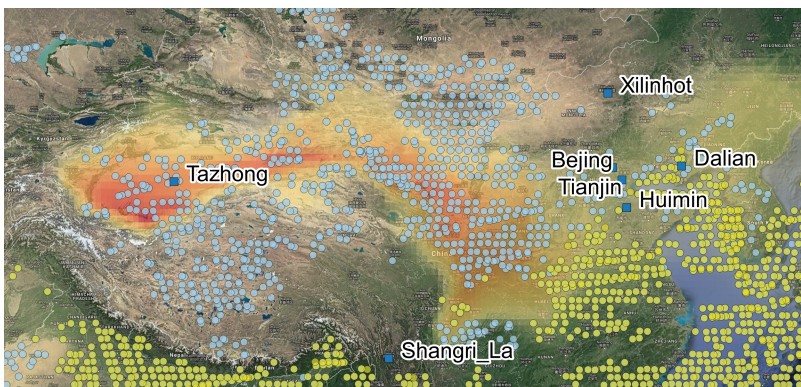

**Figure 1.** Desert Dust event forecast by the CAMS system for the 9th of March 2013 (shaded) with observation locations from the Dark-Target (yellow dots) and the deep-blue algorithms (blue dots) from MODIS Collection 6 used in the assimilation experiments. A few in situ verification stations are also highlighted. For these stations detailed results are provided.

    Figure 3 shows a first look at the model performance via 2D histograms of model versus observed values of AOD at 440 nm for March-May 2013. All stations are considered together. Data are plotted for central/northern China (30N-45N, 75E-135E). Visual inspection of the plots shows that both assimilation experiments have a more similar distribution than the control

experiment with respect to the observations, particularly for large dust events (red colors). The mean of the observations is 0.593 versus 0.481 for experiment CONTROL, 0.599 for experiment DT and 0.602 for experiment DTDB. Correlation is the highest for experiment DTDB at 0.725 versus 0.707 for experiment DT and 0.668 for experiment CONTROL.

    The statistical indicators presented here are Fractional Bias (FB, sometimes also referred to as Modified Mean Normalized Bias) and Fractional Gross Error (FGE), although other indicators have been analyzed and have been found to lead to similar

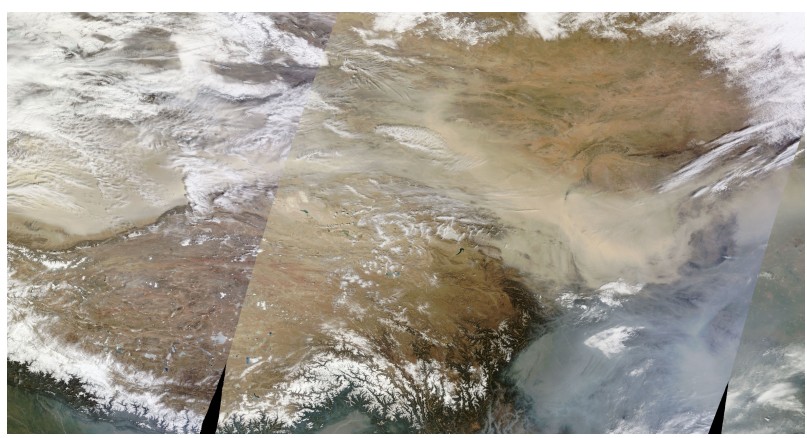

**Figure 2.** NASA Worldview image for March 9, 2013. Available at https://go.nasa.gov/2FI0vbI.

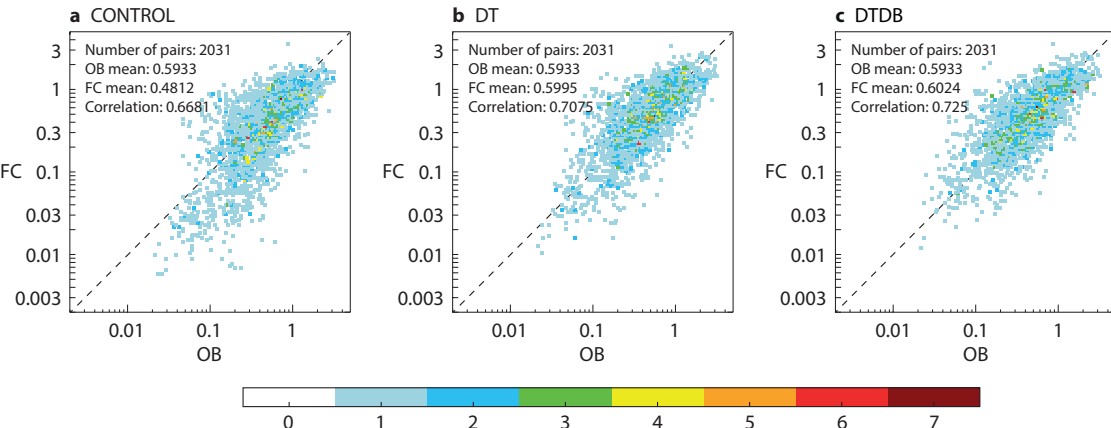

**Figure 3.** Two-dimensional histogram of modeled versus observed AOD at 440 nm for all stations (33 in total) in central/northern China (30N-45N, 75E-135E). Panel (a) shows experiment CONTROL, panel (b) shows experiment DT and panel (c) shows experiment DTDB. FC stands for forecast (model simulation) and OB stands for observations.

conclusions. The mathematical definitions for these parameters are the following:

$$FB = \frac{2}{n} \sum_{i=1}^{n} \frac{f_i - o_i}{f_i + o_i} \tag{1}$$

$$FGE = \frac{2}{n} \sum_{i=1}^{n} \frac{|f_i - o_i|}{|f_i + o_i|} \tag{2}$$

where $f_i$ is the model output and $o_i$ are the observations. $n$ represents the total number of observations. The FB is normalized to make it non-dimensionless. It varies between $+2$ and $-2$ and has an ideal value of zero for perfect model (and perfect observations). The FGE ranges between $0$ and $2$ and behaves symmetrically with respect to under- and overestimation, without over-emphasizing outliers (as instead the root mean square error does). See Yu et al. (2006) for further information.

Figure 4 shows maps of FB for Aerosol Optical Depth (AOD) at 440 nm with respect to all AERONET and CARSNET sites for March-May 2013 over central/northern China at the analysis time. Within a certain margin of noise, it is possible to see that the performance of both experiments including assimilated MODIS data is more skillful than that of the control experiment. The figure also shows that including the Deep Blue data, which provide Aerosol Optical Depth information over

bright surfaces, helps to further reduce the bias with respect to the ground-based observations. Figure 5 shows maps of FGE and confirms that the error with respect to the independent observations is reduced in the runs with assimilated satellite data.

Time-series of FB and FGE average over the same area are shown in figure 6. Again, the improvement in bias and error in the assimilation experiments is visible. The model has generally a similar bias when compared to the CARSNET observations and AERONET stations over this region. The bias is particularly large for the CONTROL experiment, except for some days.

Yet again the experiments with assimilated satellite data perform visibly better than the CONTROL. If we focus on the FGE, its maximum value during the study period reached values of 1. Focusing for example on April 10, the error in the experiment CONTROL is close to 1.1 with respect to the AERONET stations (solid red line) and 0.7 with respect to the CARSNET station (dotted red line). However, the FGE for the DT and DTDB experiments is around 0.2 for AERONET stations (green and blue solid lines) and 0.4 for CASRNET stations (green and blue dotted lines) with a 50-80% reduction in error due to assimilation of

satellite data. However, on other dates, the improvement is less marked. The dust storm case of March 9th is quite interesting. The FGE is 0.9 for all experiments with respect to the CARSNET data (red, green and blue dotted lines), but is reduced to 0.3 the following day with a faster decrease for the DTDB experiment (blue dotted line) than for the other experiments. This indicates the additional benefits of including observational satellite information over source areas. Curiously on March 10th one can observe a similar behavior but this time with respect to the AERONET observations with FGE going from 0.9 in

the CONTROL (red solid line) to 0.4 in experiments DT and DTDB (green and blue solid lines) for an improved agreement with the AOD at the AERONET stations close to 60%. In this case, the satellite data have improved the analysis observations downwind of the storm (see figure 1).

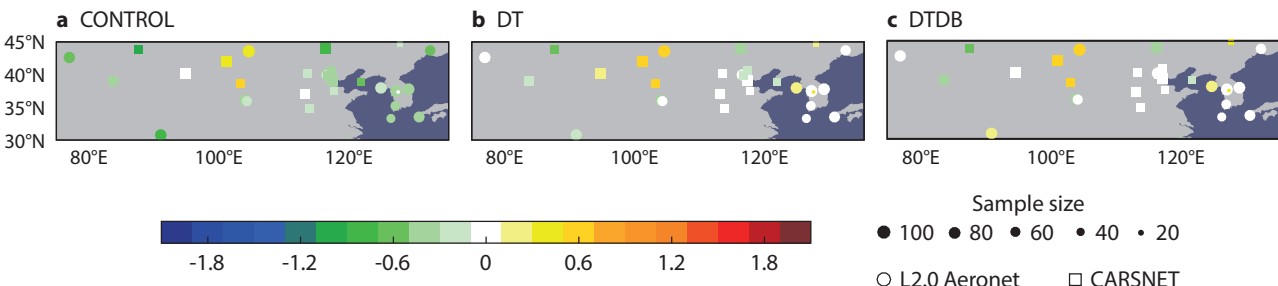

**Figure 4.** Fractional Bias for Aerosol Optical Depth at 440nm with respect to AERONET (circles) and CARSNET (squares) data: panel (a) shows the experiment CONTROL, panel (b) shows experiment DT (with only Dark Target MODIS retrievals in the assimilation), and panel (c) shows experiment DTDB (with both Dark Target and Deep Blue MODIS retrievals in the assimilation).

Specific sites in China have also been selected in order to understand the model skill in both remote and urbanized regions that are particularly affected by Asian dust plumes. The agreement between the experiments and independent observations is

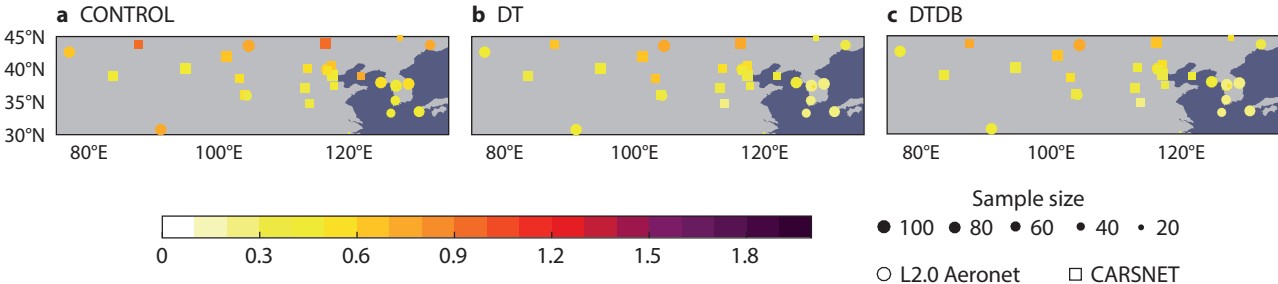

**Figure 5.** As in for figure 4 but for Fractional Gross Error.

generally better for AERONET stations than CARSNET stations with generally lower NMB and FGE, highlighting the fact that the arid Asian regions where some of the CARSNET stations are located are more challenging for the model to represent correctly.

This can also be seen in Figure 7 where the model is compared with the observations at several CARSNET stations over the the whole spring season 2013. The increase in AOD corresponding to the passage of the 9th of March storm shown in figure 1 is evident in the ground-based CARSNET observations. The magnitude of the dust peak in the source region is best captured by the runs with assimilated Deep Blue AOD data, even though the secondary dust peak is still underestimated by the model, as shown in the time-series for Tazhong (39.0N-83.67E). The improvement coming from assimilation is especially felt downwind of the source regions, as seen for example in Shangri-La (28.03N-99.73E) to the South-East, and Xilinhot (43.95N-116.12E) to the North-East.

## 4.2 Impact of the assimilation on the short-range AOD forecasts

In the previous subsection, results have been presented that show how assimilation of AOD satellite data from the MODIS retrievals brings beneficial improvements to the simulation of AOD within the ECMWF/CAMS system. The impact on the analysis is clearly shown in the improvement in the FB and FGE. The question that might arise at this point is whether the impact of the assimilated satellite data is only felt at the analysis time or whether it carries on to the subsequent forecast. To answer this question, the spatial data relative to FB shown in figure 4 are summarized in figure 8 using an histogram which shows the FB as a function of the number of AERONET and CARSNET sites. The upper panels show the histogram for the analysis time, whereas the lower panels show the same for the 48h forecast. Again, we can observe that the skill of the experiments with data assimilation is higher with respect to the control experiment. While this increased skill is higher at analysis time, it persists into the short-range forecast at 48h.

Figure 9 shows the behavior of FB and FGE as a function of forecast range. This is another way to display what already shown in figure 8. It can be seen that the average FB (FGE) over central-northern China is lower of about 0.4 (0.2) for experiments DT and DTDB with respect to the control experiment. Of particular interest is the fact that the skill of the two

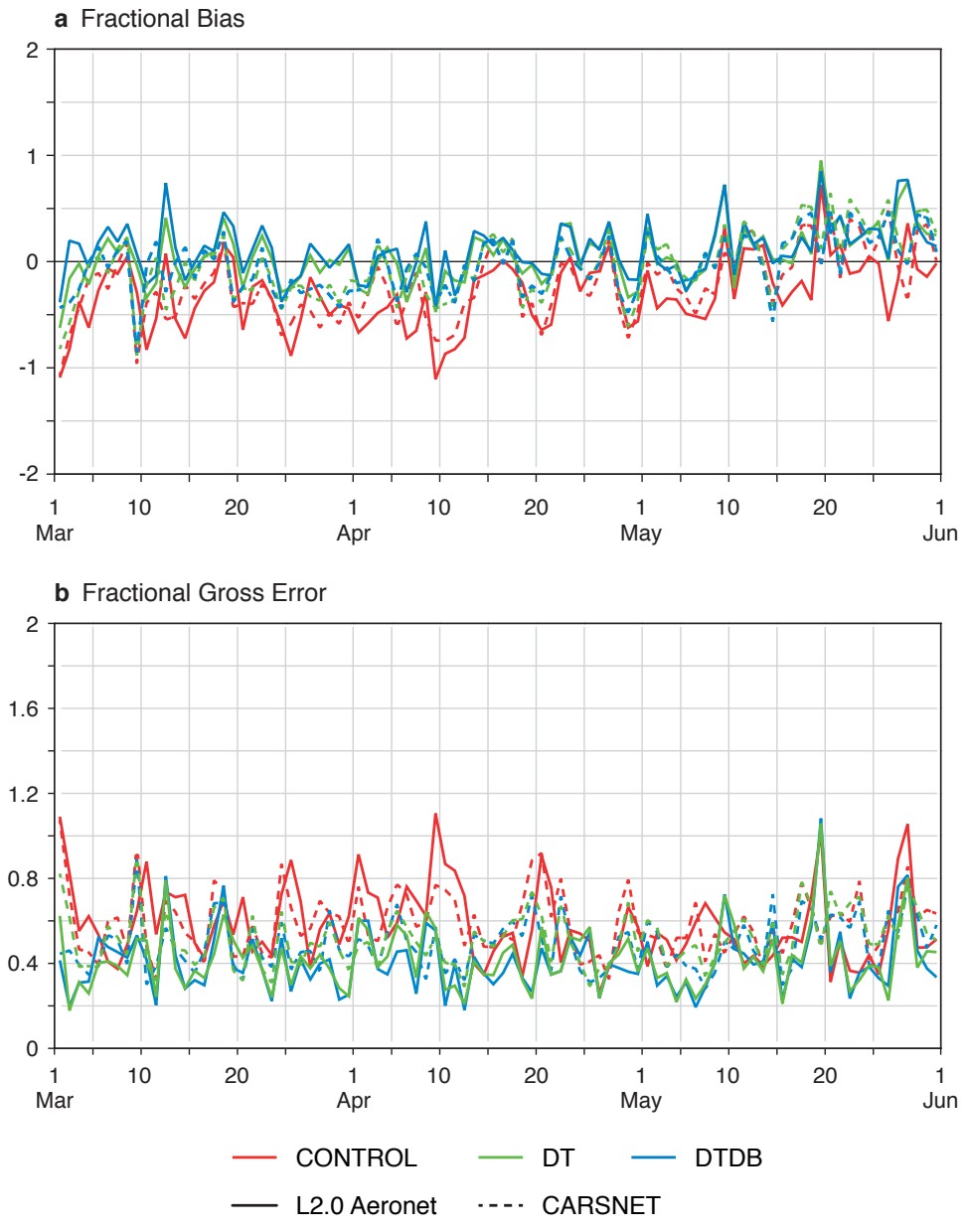

**Figure 6.** Time-series of Fractional Bias (a) and Fractional Gross Error (b) in AOD at 440nm averaged of the central/northern China area (33 sites). Solid lines are relative to AERONET data whereas dotted lines are relative to CARSNET data. Red is experiment CONTROL, green is experiment DT and blue is experiment DTDB.

experiments with data assimilation is almost constant at various forecast lead times, indicating that the positive impact of the observations is also felt after the analysis into the short-range forecast.

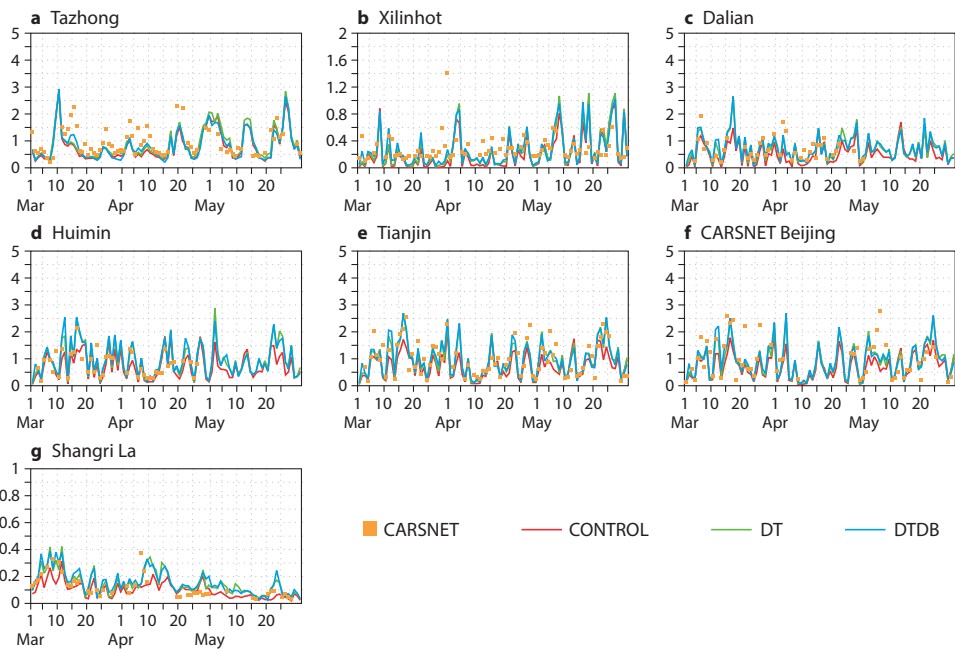

**Figure 7.** Comparison of CARSNET Aerosol Optical Depth data (blue squares) with model Aerosol Optical Depth at several CARSNET stations highlighted in 1 during the period March-May 2013. Red corresponds to experiment CONTROL, green to experiment DT, and blue to experiment DTDB. From top left to bottom right: Tazhong (39.0N-83.67E), Xilinhot (43.95N-83.67E), Dalian (38.90N-121.63E), Huimin (37.48N-117.53E), Tianjin (39.10N-117.17E), Beijing (39.80N-116.47E), Shangri-La(28.02N-99.73E).

ECMWF/CAMS forecasts are issued out to day 5. However, some users utilize of the forecast at day 2 for their applications. For example, for the management of solar power plants, it is the 48h aerosol forecast that is considered to plan the plant operations Schroedter-Homscheidt et al. (2013)). In these instances, a reduction in the forecast bias at day 2 is a welcome development. Moreover, experiment DTDB, which included both MODIS Dark Target and Deep Blue data, shows a slightly
5    lower fractional gross error in the comparison with CARSNET data than the DT experiment, indicating an additional beneficial impact of observations close to the dust sources.

## 4.3   Comparisons with independent PM data

PM data were obtained for the months of March-May 2013 from CEPA for several stations in the Beijing area. For clarity purpose only the month of March is shown in figure 10 which presents the comparison between model PM10 values and
10    observations (panel a) and a map with the location of the stations (panel b). The data for all stations show a peak in PM10 (all particles with diameter smaller than 10 $\mu m$), corresponding to the passage of the dust front on March 9. The box-plot represent the histograms of hourly data obtained from the 11 air quality stations available in the Beijing municipality. Outliers are indicated with a dot. The coarse resolution of the ECMWF model means that all station points are included in the same grid box.

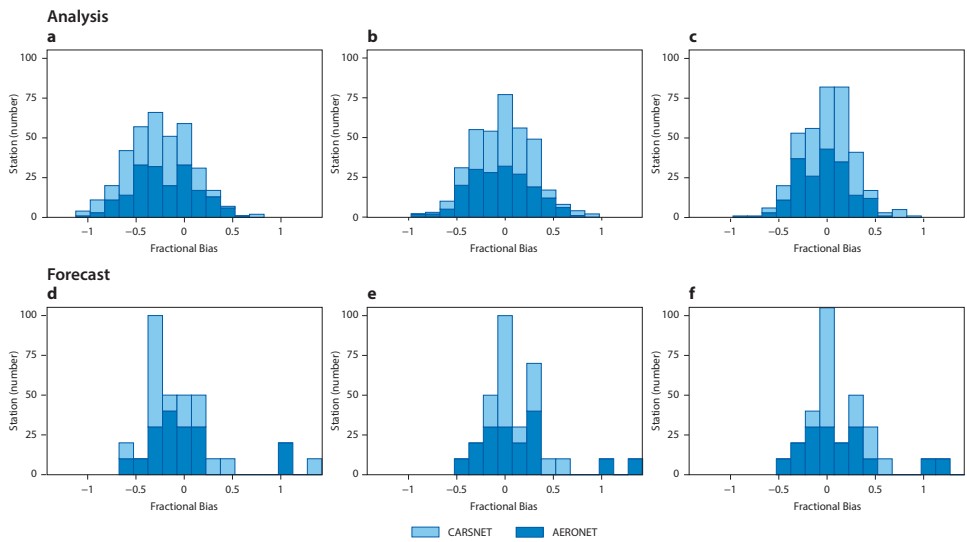

**Figure 8.** The histograms on the top and bottom rows are constructed using data from all stations displayed in the map of figure 4 and are valid at the analysis and 48h forecast times, respectively. Panels (a) are relative to the experiment CONTROL, panel (b) to the experiment DT, and panels (c) to the experiment DTDB.

The model captures the front passage which is visible in the increase of PM10. However, the timing of the PM maximum in the model is earlt by approximately 3 hours. If we consider the average value of the PM10 observation peak, then the model underestimates it by $\approx 100\ mg/m^3$. Outside the peak, the values of PM in the models are too large, indicating a broader spatial extent of the aerosol plume in the model with respect to what was observed. It is important to notice that the coarse model
resolution does not allow to resolve the fine details of the local pollution, particularly in regions in which the topography is varied. Moreover, not all species are modeled in the system. For example nitrates are not included in the model version used in this study and this missing component might partially be the cause of the clear disagreement between model and observations (see for example Wu et al. (2017)). However in situations dominated by large-scale synoptic events such as the case for the dust storm of March 9, the model can give a qualitative indication about the cause of the increased pollution and can be used to
provide a warning to the population. This is what is seen for all experiments in figure 10a. What we can also observe is that the assimilation is not able to improve the PM10 forecast. The small differences between the experiments with assimilated satellite data and the control experiment are well within the large differences between model and observations. PM10 is increased in the DTDB experiment thanks to the additional AOD data, but it does not reach the observed peak. The spurious secondary peak of March 10, not reported in the PM data, is reduced in the DTDB experiment. However, once again, the difference with
the observed values of PM is of two orders of magnitude.

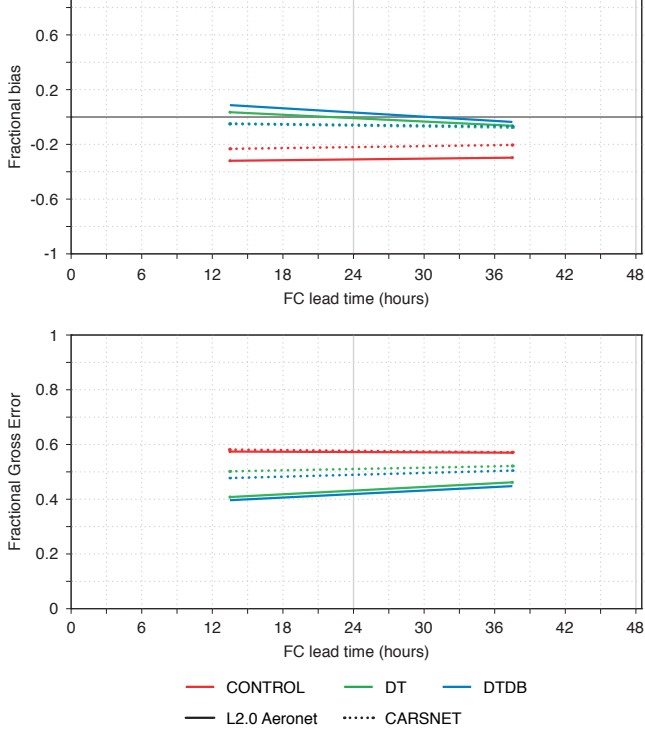

**Figure 9.** Fractional bias (top) and fractional gross error (bottom) as a function of forecast range for experiment CONTROL (red), experiment DT (green), and experiment DTDB (blue). Data are averaged over central/northern China (30N-45N, 75E-135E) for the period March-May 2013.

## 5   Conclusions

The operational ECMWF/CAMS system was used to demonstrate the impact of assimilating satellite AOD data for an improved characterization of Asian dust storms. Three experiments were conducted: one control experiment with no aerosol data assimilated, one experiment with only Collection 6 MODIS Dark Target AOD retrievals, which do not offer coverage over
5   bright surfaces and one experiment with all available MODIS data including Deep Blue AOD retrievals.

Several concluding points can be learned from these experiments. Results confirm that assimilation of advanced satellite data is effective for improving the representation of desert dust distributions over Asia and for monitoring and forecasting dust episodes (in the short-range). In particular, MODIS Deep Blue data are beneficial both in source regions (i.e. Tazhong) as well as downwind of desert areas (i.e. Xilinhot and Shangri-La). The fractional bias and the gross error with respect to independent
10   AOD observations from the AERONET and CARSNET ground-based networks are lower in the assimilation experiments than in the control forecast with no assimilated aerosol data. Timeseries of FGE averaged over central/northern China, reveal that in some extreme cases the FGE is reduced by as much as 60-80% in the experiment which includes all MODIS data (DTDB)

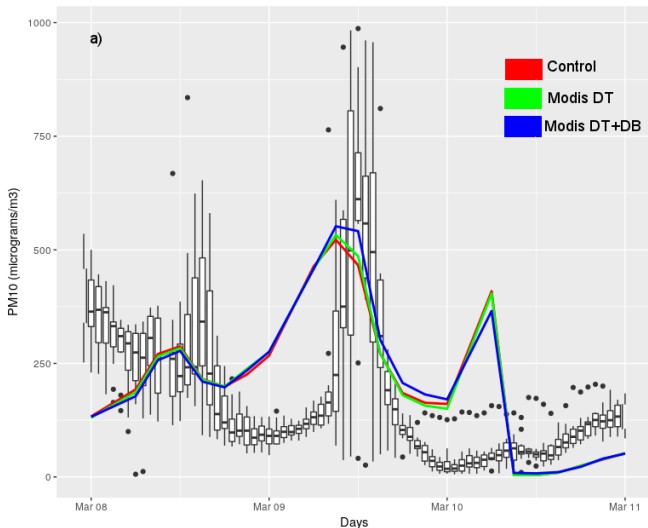
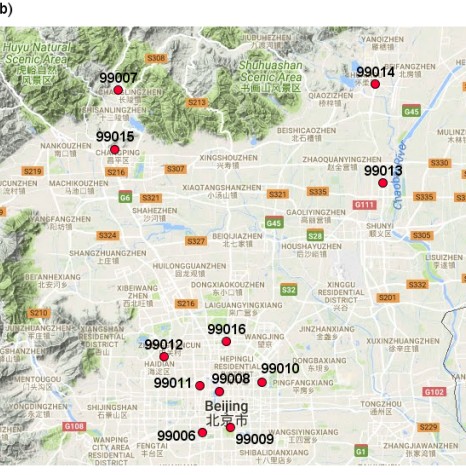

**Figure 10.** a) March 2013 PM10 time series for the measurements recorded in the 11 stations available in the Beijing (39.9N,116.4E) municipality and the simulations performed with the ECMWF model in the different configurations (CONTROL in red; DT in green and DTDB in blue). b) location and identification code of the observation stations around the Beijing area. The map covers an area of approximately $40km^2$.

with respect to the experiment which has no aerosol data (CONTROL). The impact of the assimilated data is mainly felt at the analysis time but remains visible in the 48h forecast.

Comparisons with PM10 data from the CEPA network for March 2013 in the Beijing area show that the model has skill in predicting the passage of a dust front. However there is a mismatch in PM10 values with large biases (up to two orders
of magnitude in some cases) displayed by the model, even for the assimilation experiments. The assimilation of satellite AOD in this case can only make small adjustments to PM10 but is unable to improve fundamentally the quality of forecast. Despite this, the passage of the storm is captured, indicating that the global model is somewhat able to capture the regional pollution flow even at coarse resolution. This indicates that the global model analyses may be used as boundary conditions for regional air quality models at higher resolution, enhancing their performance in situations when part of the pollution may
have originated via synoptic flow mechanisms. However, the skill of the global model for PM10 is poor. This is likely due to model biases, coarse resolution, unresolved topography, lack of resolved local emissions and lack of observations to constrain the aerosol speciation and vertical structure. It is worth mentioning that AOD is less affected by these factors as it is a column-integrated variable. Also, spatial patterns of AOD do not always match with those of surface PM due to variability in aerosol hygroscopicity and relative humidity which can affect AOD greatly but leave PM10 completely unchanged.
These results highlight the crucial importance of the verifying datasets to assess model skill and assimilation impact. CARSNET data offer unprecedented coverage over remote Asian dust regions, which are otherwise poorly observed. From the regional distribution maps of figure 4 it is possible to see that CARSNET and AERONET stations over Asia are largely com-

plementary. Data availability in these regions both for assimilation and model evaluation and development is key to progress in further addressing the challenges of Asian dust prediction.

As part of CAMS, ECMWF is providing twice-daily forecasts of atmospheric composition (including desert dust) up to 5 days ahead, which are publicly available. These forecasts rely on assimilation of MODIS Collection 6.1 AOD (both Dark Target and Deep Blue data) as well as PMAP data. They have been shown to provide useful information and reliable boundary conditions for local air quality applications in Europe, China and East Asia, provided they are used in conjunction with high-resolution regional air quality models.

*Acknowledgements.* This work has been initiated and partly funded by the PANDA European Project (FP7, n. 606719). The work has also been supported by the Copernicus Atmosphere Monitoring Service, which is part of the European Union's flagship space programme Copernicus. CAMS data are freely available from http://atmosphere.copernicus.eu. The data providers contributing to the AERONET and CARSNET networks are also gratefully acknowledged. We acknowledge the use of imagery from the NASA Worldview application (https://worldview.earthdata.nasa.gov/) operated by the NASA/Goddard Space Flight Center Earth Science Data and Information System (ESDIS) project. PM10 data were courtesy of the China Environmental Protection Agency network (China). The co-Editor Federic Fierli and one anonymous reviewer are gratefully acknowledged for their useful comments which have contributed to improve greatly the original version of the manuscript.

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
