# Peer review of "The value of satellite observations in the analysis and short-range prediction of Asian dust"

_Atmospheric Chemistry and Physics, 2018_

## Referee Comment (RC1) · Anonymous Referee #2 · 22 Jun 2018

This paper assesses the impact of assimilating satellite AOD on aerosol forecast of AOD and surface PM10 with the ECMWF/CAMS global system. This is done by evaluating results from three model experiments with ground-based network measurements of AOD and PM10 for March 2013: (1) free model run (CONTROL), (2) model assimilation of MODIS Dark Target AOD (DT), and (3) model assimilation of MODIS Dark Target and Deep Blue (DTDB).

The paper is well-written and well organized and the information is interesting. However, I have several major concerns/comments that should be addressed in the revision.

[Figure]

1. The title of this paper explicitly indicates that the paper is to address the impact of data assimilation on the prediction of Asian dust. However, after reading the entire paper, I feel that the paper is not particularly focused on dust forecast, and only one dust episode is shown. The evaluation of AOD with AERONET is done within an extensive region and large fraction of the area has limited influence from dust. If choosing 1020 nm AOD is for its better representation of dust, it should be clearly stated in the paper. 2. In the abstract it is said that the model experiments were run to understand the relative contribution of Asian dust to air quality over China, but there is no any results or discussion on this topic in the paper. 3. There is not enough data used in the paper (only one month) to generate robust statistics. It is stated in the introduction that the model experiments were run for one year, but only March results are used in the paper. We know that the dust season in China lasts more than just a month in March, why not using multiple months to have more data for statistics? I also noted the statement that "ECMWF is providing twice-daily forecasts of atmospheric composition (including desert dust) up to 5 days ahead", so potentially there is a lot of results to use. 4. The impact of assimilation to surface PM10 should be much better and more quantitatively evaluated. From Figure 9, it is clear that PM10 from the three model experiments are nearly identical and the satellite AOD assimilation brings little improvements of PM10 prediction. Although DTDB is seen to be a little closer to the observations at some time steps, the so-called "improvements" are practically negligible and do not change the forecast skill at all. Please provide quantitative evaluation in this case, including peak values and timing, bias, correlation, etc. that can show the difference among the three model experiments and between model runs and observations to really understand the magnitude of "better agreement". 5. Actually, the PM10 case is a very interesting one that warrants a more in-depth analysis. In the three-day simulations shown in Figure 9, what are the AOD time series look like, compared with AERONET (and/or CARSNET) AOD in the Beijing municipal area? Does AOD and PM10 vary together or not? Can you explain the AOD-PM10 relationship in terms of aerosol vertical profile, composition, and other factors (e.g., hygroscopic growth of aerosols)? What do

the results tell us about model characteristics and the effectiveness of AOD assimilation for PM10 forecast? 6. The assessment needs to be more objective, more robust, and more quantitative. For example, within the year of 2013, how many days of heavy dust episodes the CONTROL experiment would miss but DT or DTDB would capture? How significant improvements the assimilation brings in heavy dust (or pollution) cases and in background cases?

Minor comments:

Page 2, line 3: Add Taklimakan as a desert of dust source.

Page 2, line 18: Typo and incomplete sentence "since 2005ch is ."

Page 4, line 6: Is the prescribed dry deposition velocity particle size dependent? Does it depend on seasons and locations?

Page 4, line 7, sedimentation: This is strange - you could argue that the errors might be insignificant for the two smaller size groups from ignoring sedimentation, but using a fixed settling velocity is not justified, since the air density and viscosity changes spatially and temporally.

Page 4, line 10: "bulk parameterization" is for particle size, right?

Page 4, line 12-13, "Removal processes include sedimentation of all particles": This sentence directly contradicting with the sentence in line 7 that "sedimentation is applied only to the largest dust bin".

Page 4, line 14: How is sulphate formation from SO2 is dealt with in the model?

Page 4, line 23: What "atmospheric composition variables" are assimilated that are relevant to this study?

Page 4, line 24-26: How do you deal with the aerosol hygroscopic growth? How do you factor that in when you redistribute the aerosol mixing ratio at the end of minimization?

[Figure]

Page 4, line 19-30, vertical profile: Please make it clear that the vertical profiles are all from the model; no data assimilation for aerosol vertical profiles.

Page 5, line 32: Change "1" to "Figure 1".

Page 6, line 1 and Figure 1 and 2: The different spatial domains between Figure 1 and 2 makes it hard to visually relate the dust plume locations. I suggest make these two figures for the same geographic area or mark the Figure 2 area on Figure 1.

Page 6, line 3: From Figure 1, it looks that the dust storm originated in Taklimakan.

Page 6, line 4: transported to southeast, instead of southwest?

Page 6, line 7: Are the observed values from AERONET and CARSNET? What is the reason for using AOD at 1020 nm instead of 550 nm MODIS retrieved?

Page 6, line 8-9, SE Asia are: This is a large area. The stations within this area must have quite different aerosol composition. How many of them are surely being impacted by dust in your analysis?

Page 6, line 11-12: Four-digit after the decimal seems an over kill and means little. The differences are small: R = 0.74, 0.75, and 0.76. To what degree it matters? What are the RMSEs for these cases?

Page 7, Figure 3: What do the different colors represent?

Page 7, line 6-7: In the case of dust storms or episodes the "outliers" are probably the most critical ones for measuring the model skill.

Page 8, line 8-9: Can you quantify the model agreement with CARSNET and AEROENT separately? Is there any collocated CARSNET and AERONET stations to compare the differences? Do they use the same type of instrument? What are the known uncertainties of their instruments? Any calibration issues?

Page 9, line 5: Change "AD" to "AOD"

[Figure]

Page 9, line 10: Where is the summary given? Figure 8?

Page 9, Section 4.2: It would be informative to know after how long the benefit of data assimilation disappears, and what does it tell us about the importance of the quality of the model itself.

Page 10, line 5-8: Too many subjective statements here. How much off is the timing that is "slight wrong"? What is the standard for "good agreement" (e.g., within x%)? What is the measure of the model skill that warrants the achievement of "a good degree of skill"? The evaluation is too descriptive and not quantitative.

Page 10, line 8-10: "the experiments with assimilated satellite data draw closer to the observations": How much closer? 1%, 5%, or 50%? The three lines in Figure 9 are nearly identical and I am not sure what matrix you use to benchmark the improvements? Clearly, quantitative assessment is needed. Can you use R, FB, and FGE for assess the results of PM10 here, similar to what you did for AOD, in order to quantitatively measure the effectiveness of assimilating satellite AOD on predicting PM10 concentrations?

Page 10, line 10 (continued on Page 12 line 1), the "spurious secondary peak of March 10": DTDB is about 370 ug/m3, which is probably 20 ug/m3 lower than CONTROL, but still more than 300 ug/m3 higher than the observation! It is hard to mark it as improvement.

Relative question regarding Figure 9: It would be helpful to indicate the MODIS overpass time that the data are ingested in the assimilation system. Clearly, the nearly identical time series of the three model runs reflect the fundamental characteristics of the model processes, of which the satellite data assimilation is not able to change. The opposite diurnal variations between data and model do not change at all, the more than 2x over estimation from late Mar 08 to mid Mar 09 remains the same magnitude among the three model runs, and the model behavior in late mar 09 to Mar 10 does not change at all from CONTROL to DTDB after the strong dust episode in Mar 9. So

what have we learned from it? To me, the figure has told me that the assimilation of satellite AOD (1 or 2 time/day?) in this case helps make small adjustment of PM10 but is unable to change the quality of forecast.

Page 11, Figure 7: Please show statistics of the comparisons at each site. What are "gmsy", "goij", and "goik" in the legend?

Also, a general comment on the color scheme: model runs of CONTROL, DT, and DTDB are represented in green, red, and blue in Figure 6, but green, red, and gray (dashed line) in Figure 7, and yet, they are red, green, and blue in Figure 9! Please keep the color scheme and style consistent.

Page 12, Figure 8 caption: There are only two rows in Figure 8 and there is no "middle" row.

Page 12, line 9-10: As I mentioned again and again, the effectiveness of assimilating satellite data needs to be quantitatively assessed. The assessment of the impact on daily AOD (not just for dust) forecast is more quantitatively done, but the assessment of the impact of diurnal variation of PM10 is mostly addressed by visual impression and subjective.
* * *

---

## Editor Comment (EC1) · F. Fierli (Editor) · 18 Aug 2018

Dear Authors,

Due to the unexpected difficulty to obtain a second independent review, I will provide my evaluation of the paper as Editor / Reviewer. Such delays doesn't occur often and I should apologize for the time needed to close the discussion of your paper. Thanks for your patience.

I have read carefully the paper and, in agreement with reviewer 1, I find it highly interesting, well written and with a high pertinence to ACP scopes.

I have nevertheless a major issues that relates to the methodology of evaluation.

(1) The comparison is limited to the month of March 2013 while MODIS data shows that April and May 2013 have high levels of AOD (dust) in the region. So it would be highly desirable to extend the period of analysis and provide a more structured presentation of the events that are included in the analysis period in addition to the snapshot of figure 1. This may improve the quality of the analysis as outlined below.

(2) Moreover, a weakness is the demonstration of the improvement of forecast that is central to your analysis. I am convinced that limited additional work may improve the quality of the paper and its potential impact.

Few specific points:

- Figure 4 aims at showing a bias reduction. This is qualitatively discernible from color scale. Nevertheless, it would be more convincing to propose a quantitative table in addition to the time series of figures 6-7. A comment on why the spatial distribution of the improvement is desirable.

- While comparison with AERONET provides convincing outcomes, the one with CARSNET, as also stated by the authors is less discernible. Especially the expected increase in AOD related to the March 9th event that is almost not visible in the observations of figure 7. It is difficult to say whether improvement is achieved or not (e.g. look for instance at Tahzong site)

- A set of model maps from three experiments may be desirable here to evaluate where and how assimilation improves / modify the results.

- The method of comparison with the independent data may be better outlined. It is mentioned that all stations in a single grid box are considered. Are they aggregated ? averaged ?

- The effect of assimilation in forecasts, as expected, is less evident and limited in this analysis to histograms of figure 8. This is just partly evident from comparing fractional bias in the bottom line. A more quantitative statement would be desirable here. Moreover, also FGE would be interesting. It would also be useful to have time series. As said above, it would be much beneficial to extend the period of analysis.

- The improvement described in figure 9 for the Beijing area is not particularly striking. The authors states that higher values (less than 10%) of PM10 for experiment Modis DT+DB are due to the benefit of assimilating additional data.

- Conclusions may report a more through discussion on the system skills and limitations before stating in the last sentence about the "usability" of AOD forecasts

Minor (editorial) issues:

Page 2 line 18:
"2005ch is" → Typo ?

Page 3 line 23: please detail what CEPA is

Page 4 line 20: add the CAMS website where data are freely available. You may state it also in the acknowledgements

Page 5, 15: A sentence to clarify why the experiments where carried out at 80 km instead of 40 km and an assessment of the validity of results for both resolutions would be useful here

Page 9, 14: Sentence on Users and forecast time is not clear (it seems obvious that an improvement at T0+48 is useful) - Clarify – extend or skip it.

Page 9, Section 4.2: I guess the discussion refers to figure 8 – please refer to it.

Figure 7: please, use the same terminology for experiments in the caption

Figure 9: same as above

---

## Author Comment (AC1) · 18 Oct 2018

Reply to Reviewer #1:

Many thanks to the reviewer for all the useful comments. Please find our point-by-point answers below.

**1. The title of this paper explicitly indicates that the paper is to address the impact of data assimilation on the prediction of Asian dust. However, after reading the entire paper, I feel that the paper is not particularly focused on dust forecast, and only one dust episode is shown. The evaluation of AOD with AERONET is done within an extensive region and large fraction of the area has limited influence from dust. If choosing 1020 nm AOD is for its better representation of dust, it should be clearly stated in the paper.**

*We take the reviewer's point and hence changed the title of the paper to "The value of satellite observations in the analysis and short-range prediction of Asian dust" which reflects better the content and the scope of the paper. As far as wavelength for verification, there is no 550 (or 500nm) in the CARSNET data, that is why the 1020nm was initially chosen (for selected statistics). In the revised version of the paper now we consistently use the 440nm for all verification measures.*

**2. In the abstract it is said that the model experiments were run to understand the relative contribution of Asian dust to air quality over China, but there is no any results or discussion on this topic in the paper.**

*We agree with the reviewer in his/her critique of the paper. The abstract has been changed to clarify that the focus is on aerosol optical depth monitoring rather than air quality.*

**3. There is not enough data used in the paper (only one month) to generate robust statistics. It is stated in the introduction that the model experiments were run for one year, but only March results are used in the paper. We know that the dust season in China lasts more than just a month in March, why not using multiple months to have more data for statistics? I also noted the statement that "ECMWF is providing twice-daily forecasts of atmospheric composition (including desert dust) up to 5 days ahead", so potentially there is a lot of results to use.**

*Statistics for three months (March-May) are now presented for the AOD verification.*

**4. The impact of assimilation to surface PM10 should be much better and more quantitatively evaluated. From Figure 9, it is clear that PM10 from the three model experiments are nearly identical and the satellite AOD assimilation brings little improvements of PM10 prediction. Although DTDB is seen to be a little closer to the observations at some time steps, the so-called "improvements" are practically negligible and do not change the forecast skill at all. Please provide quantitative evaluation in this case, including peak values and timing, bias, correlation, etc. that can show the difference among the three model experiments and between model runs and observations to really understand the magnitude of "better agreement".**

**5. Actually, the PM10 case is a very interesting one that warrants a more in-depth analysis. In the three-day simulations shown in Figure 9, what are the AOD time series look like, compared with AERONET (and/or CARSNET) AOD in the Beijing municipal area? Does AOD and PM10 vary together or not? Can you explain the AOD-PM10 relationship in terms of aerosol vertical profile, composition, and other factors (e.g., hygroscopic growth of aerosols)? What do the results tell us about model characteristics and the effectiveness of AOD assimilation for PM10 forecast?**

**6. The assessment needs to be more objective, more robust, and more quantitative. For example, within the year of 2013, how many days of heavy dust episodes the CONTROL experiment would miss but DT or DTDB would capture? How significant improvements the assimilation brings in heavy dust (or pollution) cases and in background cases?**

*We take the above points made by the reviewer. It is true that the model has high predictive skill in Aerosol Optical Depth, but low predictive skill when it comes to surface concentrations. This in it itself would warrant a more in-depth study. However, within the scope of the current paper, we have restricted ourselves to study the impact of the assimilation of satellite data for the monitoring and short-range prediction of aerosol load as described by the optical depth. The model skill is quantitatively shown to be improved for aerosol optical depth when satellite data are assimilated by comparing with ground-based, independent, measurement of aerosol optical depth. However, the picture is not so clear-cut for the surface concentrations. The global model, run at 80km resolution, is not capable to resolve local pollution features and the complex topography that, particularly in China, is responsible for pockets of extreme values of particulate matter. In a sense, using this model to address the prediction of PM10 is an ill-posed problem. We prefer hence to only show a qualitative assessment of the PM10 which shows an indication of the potential of the model in identifying large synoptic events; we discuss the limitations of the use of a coarse-resolution model to provide prediction of local pollution; and we suggest the use of the global model to provide boundary conditions for high-resolution regional models that can provide a more accurate depiction of the particulate matter. Considering a re-focusing of the paper, the comments of the reviewer are addressed below.*

**Minor comments:**

**Page 2, line 3: Add Taklimakan as a desert of dust source.**
*This was added.*

**Page 2, line 18: Typo and incomplete sentence "since 2005ch is ."**
*Typo was corrected*

**Page 4, line 6: Is the prescribed dry deposition velocity particle size dependent? Does it depend on seasons and locations?**
*Yes, the dry deposition is size dependent - it is parameterized as a modification of the instantaneous surface flux by what comes down from the layer just above the surface. In that sense, it does vary temporally and spatially.*

**Page 4, line 7, sedimentation: This is strange - you could argue that the errors might be insignificant for the two smaller size groups from ignoring sedimentation, but using a fixed settling velocity is not justified, since the air density and viscosity changes spatially and temporally.**
*That sentence was removed as the sedimentation is actually applied to all sizes for sea salt and dust in the model version used for these experiments.*

**Page 4, line 10: "bulk parameterization" is for particle size, right?**
*Yes, meaning that there is only one tracer representing the mass of the carbonaceous aerosols and of SO4. For desert dust and sea salt, the size information is actually represented by tracers in the three size bins.*

**Page 4, line 12-13, "Removal processes include sedimentation of all particles": This**

**sentence directly contradicting with the sentence in line 7 that "sedimentation is applied only to the largest dust bin".**
*This was in fact wrong and has been changed.*

**Page 4, line 14: How is sulphate formation from SO2 is dealt with in the model?**
*Sulphate is formed from SO2 using a parameterization based on RH and temperature following Eatough et al. (1994) and latitude following Huneeus et al., 2009.*

**Page 4, line 23: What "atmospheric composition variables" are assimilated that are relevant to this study?**
*Directly relevant to this study only the aerosol mixing ratio. That phrase was intended to be more general and to describe the CAMS system which has the capability to assimilate ozone, SO2, CO, NOx, etc.*

**Page 4, line 24-26: How do you deal with the aerosol hygroscopic growth? How do you factor that in when you redistribute the aerosol mixing ratio at the end of minimization?**

*The hygroscopic growth for sea salt is parameterized according to Tang et al (1997). The conversion from hydrophobic to hydrophilic organic matter and black carbon follows an exponential law with conversion rate 7.1E-06_JPRB, following Boucher et al (2002). This treatment is detailed in Morcrette et al (2009), referenced in the paper.*

*At the beginning of the minimization the total mass is calculated as the sum of all contributing aerosol species at that specific location. The fractional contribution of each species is maintained constant over the 12-h assimilation window. At the end of the minimization the increments on total aerosol mixing ratio are redistributed to the various species according to their fractional contribution calculated at the beginning. In a way, all aerosol physical processes are considered constant over the assimilation window and tendencies are not updated. This is of course an approximation.*

**Page 4, line 19-30, vertical profile: Please make it clear that the vertical profiles are all from the model; no data assimilation for aerosol vertical profiles.**
*This has been clarified and the following sentence has been added:*
*"The vertical profile of the aerosol mixing ratio is not modified by the assimilation as only AOD is used as observation. Thus, the vertical profile is dictated by the model."*

**Page 5, line 32: Change "1" to "Figure 1".**
*Added*

**Page 6, line 1 and Figure 1 and 2: The different spatial domains between Figure 1 and 2 makes it hard to visually relate the dust plume locations. I suggest make these two figures for the same geographic area or mark the Figure 2 area on Figure 1.**

*This has been addressed.*

**Page 6, line 3: From Figure 1, it looks that the dust storm originated in Taklimakan.**

*This has been clarified in the text.*

**Page 6, line 4: transported to southeast, instead of southwest?**
*The typo was corrected. The sentence now reads: "The storm originated in the Taklimakan and dust was first transported to Northeastern China and further transported to the southeast."*

**Page 6, line 7: Are the observed values from AERONET and CARSNET? What is the**

**reason for using AOD at 1020 nm instead of 550 nm MODIS retrieved?**

*The AOD observations are from AERONET and CARSNET. Since CARSNET does not provide observations of AOD at 550nm, the 1020nm AOD was used instead in the previous version of the paper. This has been now changed to 440nm which is closer to 550nm.*

**Page 6, line 8-9, SE Asia are: This is a large area. The stations within this area must have quite different aerosol composition. How many of them are surely being impacted by dust in your analysis?**

*The area has been reduced to central/northern China (30-45N/75-135E). This will decrease the total number of stations, but increase the number of stations which are directly impacted by dust for the season under consideration.*

**Page 6, line 11-12: Four-digit after the decimal seems an over kill and means little. The differences are small: R = 0.74, 0.75, and 0.76. To what degree it matters? What are the RMSEs for these cases?**

*We think that there are enough observations being taken into account that perhaps in this case the decimal point might be significant.*

**Page 7, Figure 3: What do the different colors represent?**

*The figure represents 2-dimentional histograms where the colour represents the number in each bin.*

**Page 7, line 6-7: In the case of dust storms or episodes the "outliers" are probably the most critical ones for measuring the model skill.**

*This is surely true but for the rest of the verification period, the more "balanced" verification metrics do more justice to the model and are perhaps more robust.*

**Page 8, line 8-9: Can you quantify the model agreement with CARSNET and AEROENT separately? Is there any collocated CARSNET and AERONET stations to compare the differences? Do they use the same type of instrument? What are the known uncertainties of their instruments? Any calibration issues?**

*We are only users of the data and provide references for the CARSNET and AERONET datasets. We are not aware of co-located stations.*

**Page 9, line 5: Change "AD" to "AOD"**

*Changed.*

**Page 9, line 10: Where is the summary given? Figure 8?**

*Yes, in figure 8. This has been added in the text for clarity.*

**Page 9, Section 4.2: It would be informative to know after how long the benefit of data assimilation disappears, and what does it tell us about the importance of the quality of the model itself.**

*This has been addressed by adding a new figure (Figure 9 in the revised version) with quantitative scores of model skill. There is evidence that up to 48 hours the benefits of satellite data assimilation are still felt in the short-range prediction. This is now discussed more in depth and for a longer verification period (March-May 2013).*

**Page 10, line 5-8: Too many subjective statements here. How much off is the timing that is "slight wrong"? What is the standard for "good agreement" (e.g., within x%)? What is the measure of the model skill that warrants the achievement of "a good degree of skill"? The evaluation is too descriptive and not quantitative.**

*We have acknowledged that the comparison with PM10 is qualitative and also that the skill of the model is low when predicting particular matter.*

**Page 10, line 8-10: "the experiments with assimilated satellite data draw closer to the observations": How much closer? 1%, 5%, or 50%? The three lines in Figure 9 are nearly identical and I am not sure what matrix you use to benchmark the improvements? Clearly, quantitative assessment is needed. Can you use R, FB, and FGE for assess the results of PM10 here, similar to what you did for AOD, in order to quantitatively measure the effectiveness of assimilating satellite AOD on predicting PM10 concentrations?**

*It has been noted that the impact of satellite data on the monitoring of PM10 is limited to situations with clear synoptic structure, but otherwise the model has little skill due to coarse resolution, missing species, and unresolved local sources.*

**Page 10, line 10 (continued on Page 12 line 1), the "spurious secondary peak of March 10": DTDB is about 370 ug/m3, which is probably 20 ug/m3 lower than CONTROL, but still more than 300 ug/m3 higher than the observation! It is hard to mark it as improvement.**

*This has been noted.*

**Relative question regarding Figure 9: It would be helpful to indicate the MODIS overpass time that the data are ingested in the assimilation system. Clearly, the nearly identical time series of the three model runs reflect the fundamental characteristics of the model processes, of which the satellite data assimilation is not able to change. The opposite diurnal variations between data and model do not change at all, the more than 2x over estimation from late Mar 08 to mid Mar 09 remains the same magnitude among the three model runs, and the model behavior in late mar 09 to Mar 10 does not change at all from CONTROL to DTDB after the strong dust episode in Mar 9. So what have we learned from it? To me, the figure has told me that the assimilation of satellite AOD (1 or 2 time/day?) in this case helps make small adjustment of PM10 but is unable to change the quality of forecast.**

*The main text and conclusions have been rewritten to highlight the weak points as discussed by the reviewer above. For example the concluding paragraph on the PM evelation now reads: "Comparisons with PM10 data from the CEPA network for March-May 2013 in the Beijing area show that the model is has some skill in predicting the passage of a dust front. However there is a mismatch in PM10 values with large biases (up to an order of magnitude in some cases) displayed by the model, even for the assimilation experiments. The assimilation of*

*satellite AOD in this case helps make small adjustments to PM10 but is unable to change the quality of forecast. Despite this, the spatial patterns are well captured and the global model is able to capture the regional pollution patterns even at the coarse resolution. This indicates that the global model analyses may be used as boundary conditions for regional air quality models at higher resolution, enhancing their performance in situations when part of the pollution may have originated via large-scale mechanisms. However, the skill of the global model for PM10 is not as good as for AOD, due to model biases, coarse resolution, lack of resolved local emissions and lack of observations to constrain the aerosol speciation and vertical structure."*

**Page 11, Figure 7: Please show statistics of the comparisons at each site. What are "gmsy", "goij", and "goik" in the legend?**

*gmsy is the CONTROL run, goij is the DT run and goik is the DTDB run.*
*This has been corrected and the experiment names do not appear any longer in the plots.*

**Also, a general comment on the color scheme: model runs of CONTROL, DT, and DTDB are represented in green, red, and blue in Figure 6, but green, red, and gray (dashed line) in Figure 7, and yet, they are red, green, and blue in Figure 9! Please keep the color scheme and style consistent.**

*This has been corrected.*

**Page 12, Figure 8 caption: There are only two rows in Figure 8 and there is no "middle" row.**
*Typo corrected.*

**Page 12, line 9-10: As I mentioned again and again, the effectiveness of assimilating satellite data needs to be quantitatively assessed. The assessment of the impact on daily AOD (not just for dust) forecast is more quantitatively done, but the assessment of the impact of diurnal variation of PM10 is mostly addressed by visual impression and subjective.**

*We now acknowledge this explicitly. The global mode is not apt at representing local PM10 features, and that is clearly shown in the qualitative comparison. However, the passage of the dust front is visible in the model so we argue that the global model can be used for boundary conditions for high resolution air quality models which would give a more accurate representation of the local pollution.*

---

## Author Comment (AC2) · 18 Oct 2018

**Reply to the Editor**

**Dear Authors,**

**Due to the unexpected difficulty to obtain a second independent review, I will provide my evaluation of the paper as Editor / Reviewer. Such delay doesn't occur often and I should apologize for the time needed to close the discussion of your paper. Thanks for your patience.
I have read carefully the paper and, in agreement with reviewer 1, I find it highly interesting, well written and with a high pertinence to ACP scopes.**

*Dear Editor,*

*We would like to thank you for your effort in seeing this paper reviewed and evaluated for possible publication, including serving as a reviewer. Please find our answers to your comments below.*

**I have nevertheless major issues that relates to the methodology of evaluation.**

**(1) The comparison is limited to the month of March 2013 while MODIS data shows that April and May 2013 have high levels of AOD (dust) in the region. So it would be highly desirable to extend the period of analysis and provide a more structured presentation of the events that are included in the analysis period in addition to the snapshot of figure 1. This may improve the quality of the analysis as outlined below.**

*The period of analysis has been extended to three months (March-May 2013) which covers the whole Asian dust season. Moreover the spatial area has been restricted to central/northern China to be able to focus better on the dust monitoring.*

**(2) Moreover, a weakness is the demonstration of the improvement of forecast that is central to your analysis. I am convinced that limited additional work may improve the quality of the paper and its potential impact.**

*The evidence for an impact in the forecast has been added (new figure showing both bias and FGE as a function of forecast range). However, the title was changed to refocus the paper. It now reads: "The value of satellite observations in the analysis and short-range prediction of Asian dust" which describes better the intention of the paper and does not arise unmet expectations in the reader.*

**Few specific points:**

**-Figure 4 aims at showing a bias reduction. This is qualitatively discernible from color scale. Nevertheless, it would be more convincing to propose a quantitative table in addition to the time series of figures 6-7. A comment on why the spatial distribution of the improvement is desirable.**

**-While comparison with AERONET provides convincing outcomes, the one with CARSNET, as also stated by the authors is less discernible. Especially the expected increase in AOD related to the March 9th event that is almost not visible in the observations of figure 7. It is difficult to say whether improvement is achieved or not (e.g. look for instance at Tahzong site)**

*Due to the change in the study area from the wider South-East Asia (23S-50N,65E-180E) initially evaluated to the current area in the revised version of the paper (Central-northern China, 30N-45N, 75E-135E), the comparison with CARSNET station data appears now clearer in the bias plots. For the station data, what is shown is now a longer time-period and it's possible to monitor the behaviour of the model over the whole spring season. In particular for Tahzong, although the agreement is far from perfect, it is possible to discern that the run with assimilation of Dark Target and Deep blue MODIS data generally outperforms the run without any aerosol data.*

**-A set of model maps from three experiments may be desirable here to evaluate where and how assimilation improves / modify the results.**

Unfortunately, the model plots do not show very well the changes. These are better captured in the maps shown in figures 4 and 5.

**-The method of comparison with the independent data may be better outlined. It is mentioned that all stations in a single grid box are considered. Are they aggregated ? averaged ?**

*This sentence was added in the text:*

*"Model data was bilinearly interpolated to the AERONET/CARSNET site locations and then averaged over 24 hour periods (from T+3 to T+24). AERONET data is similarly averaged, with each data value receiving a weight proportional to the time difference between the data values before and after it, up to a maximum of three hours. The CARSNET data used was already in the form of daily averages and no further averaging was done."*

**- The effect of assimilation in forecasts, as expected, is less evident and limited in this analysis to histograms of figure 8. This is just partly evident from comparing fractional bias in the bottom line. A more quantitative statement would be desirable here. Moreover, also FGE would be interesting. It would also be useful to have time series. As said above, it would be much beneficial to extend the period of analysis.**

*This has been done and a new figure was included in the paper to show the evolution of the fractional bias and FGE over the forecast range.*

**- The improvement described in figure 9 for the Beijing area is not particularly striking. The authors states that higher values (less than 10%) of PM10 for experiment Modis DT+DB are due to the benefit of assimilating additional data.**

*This section has completely been reworded. In particular, the value of the global model analysis and forecasts for PM10 have been criticized and put into perspective.*

**- Conclusions may report a more through discussion on the system skills and limitations before stating in the last sentence about the "usability" of AOD forecasts.**

*This has been done.*

**Minor (editorial) issues:**

**Page 2 line 18:**

**"2005ch is" → Typo ?**

*Corrected*

**Page 3 line 23: please detail what CEPA is**

*China Environmental Protection Agency, it was in the abstract but it is now repeated in the main body.*

**Page 4 line 20: add the CAMS website where data are freely available. You may state it also in the acknowledgements**

*Website added.*

**Page 5, 15: A sentence to clarify why the experiments where carried out at 80 km instead of 40 km and an assessment of the validity of results for both resolutions would be useful here.**

The resolution of the CAMS system in 2013 was 80km, and the experiments were run with that configuration. To present a comparison with the current resolution (40km) would involve running more experiments which are this point is not feasible for this study. The impact of the change in horizontal resolution on the aerosol forecasts is nevertheless an interesting point which was addressed only internally and should be the subject of a separate article.

**Page 9, 14: Sentence on Users and forecast time is not clear (it seems obvious that an improvement at T0+48 is useful) - Clarify – extend or skip it.**

*A reference with a concrete example was included to clarify the statement about the usefulness of the 48h forecasts.*

**Page 9, Section 4.2: I guess the discussion refers to figure 8 – please refer to it.**

*Reference to the figure has been added.*

**Figure 7: please, use the same terminology for experiments in the caption**

**Figure 9: same as above**

*The terminology has been homogenized.*